# DENSITY SKETCHES FOR SAMPLING AND ESTIMATION

## ABSTRACT

There has been an exponential increase in the data generated worldwide. Insights into this data led by machine learning (ML) have given rise to exciting applications such as recommendation engines, conversational agents, and so on. Often, data for these applications is generated at a rate faster than ML pipelines can consume it. In this paper, we propose Density Sketches(DS) - a cheap and practical approach to reducing data redundancy in a streaming fashion. DS creates a succinct online summary of data distribution. While DS does not store the samples from the stream, we can sample unseen data on the fly from DS to use for downstream learning tasks. In this sense, DS can replace actual data in many machine learning pipelines analogous to generative models. Importantly, unlike generative models, which do not have statistical guarantees, the sampling distribution of DS asymptotically converges to underlying unknown density distribution. Additionally, DS is a one-pass algorithm that can be computed on data streams in compute and memory-constrained environments, including edge devices.

## 1 INTRODUCTION

With the advent of big data, the rate of data generation is exploding. For instance, Google has around 3.8 million search queries per minute, amounting to over 5 billion data points or terabytes of data generated daily. Any processing over this data, such as using the training of a recommendation model, would suffer from data explosion. By the time existing data is consumed, newer data is already available. In such cases, we need to discard a lot of data. One of the critical research directions is how to reduce data storage. In this paper, we present Density Sketches (DS): an efficient and online data structure for reducing redundancy in data.

Often data comes from an underlying unknown distribution, and one of the challenges in data reduction is maintaining this distribution. In DS, we approximately store the data distribution in the form of a sketch. Using this DS, we can perform point-wise density estimation queries. Additionally, we can sample synthetic data from this sketch to use in downstream machine learning tasks. This paper shows that data sampled from DS asymptotically converges to the underlying unknown distribution. We can also view density sketches through the lens of coresets. Specifically, DS is a compressed version of grid coresets. Grid coresets are the oldest form of coresets, giving lower additive errors than modern coresets. However, grid coresets are generally prohibitive as they are exponential in dimension ($d$). DS enables us to approximate grid coresets with the dependence of memory usage depending on the actual variety in the data instead of being exponential in $d$. Also, DS provides a streaming construction for this coreset.

In this paper, we focus more on the density estimation and sampling aspects of DS. Sampling from a distribution described using data requires estimating the underlying distribution. Popular methods to infer the distribution and sample from it belong to the following three categories: 1. Parametric density estimation (Friedman et al., 2001) 2. Non-parametric estimation - Histograms and Kernel Density Estimators (KDE) (Scott, 2015) 3. Learning-based approaches such as Variational Auto Encoders (VAE), Generative Adversarial Networks (GANs), and related methods (Goodfellow et al., 2014; 2016). Generally, parametric estimation is not suitable to model most real data as it can lead to significant, unavoidable bias from the choice of the model (Scott, 2015). Learning the distribution, e.g., via neural networks, is one solution to this problem. Although learning-based methods have recently found remarkable success, they do not have any theoretical guarantees for the distribution of generated samples. Histograms and KDEs, on the other hand, are theoretically well understood. These statistical estimators of density are known to uniformly converge to the underlying true distribution *almost surely*. This paper focuses on such estimators, which have theoretical guarantees.

Storage of histograms and sampling from them is expensive because of an exponential number of partitions (also known as bins). Apart from this, histograms also suffer from the bin-edge problem: a slight variation in data can lead to significant differences in the estimation of densities. KDEs are used to solve the bin-edge problem. KDE gives a smoother estimate of density. While sampling from a KDE is efficient, KDE is expensive to store. KDE requires us to store the entire data. Coresets for KDE are a good solution to the storage problems of KDE. However, the construction of coresets is typically quite expensive. In this work, we propose *Density Sketches*(DS) - a compressed sketch of density constructed in an efficient streaming manner. DS does not store actual samples of the data. But we can still efficiently produce samples from a KDE for specific kernels, which, in turn, approximates $f(x)$. Being a compressed sketch, we can tune the accuracy-storage trade-off of DS, and we analyze this trade-off in the theorem 1.

## 2 PROBLEM STATEMENT AND RELATED WORK

**Problem Statement:** Formally, we want to create a data structure that has the following properties : (1) It sketches density information. (2) The sketch size is much smaller than the data size and does not scale linearly with it. (3) The construction is streaming and efficient. (4) We do not store any samples in the data structure created (for privacy reasons). (5) We want the sampling distribution, say $\hat{f}_S(x)$, obtained by sampling from these data structures to approximate the true underlying distribution $f(x)$.

The problem we aim to solve can be considered a data reduction problem and has been widely pursued in literature. The set of existing approaches can be broadly classified into two sections. (1) **Sampling based / Coresets** : Approaches such as clustering/importance sampling (Charikar & Siminelakis, 2017; Cortes & Scott, 2016; Chen et al., 2012) and coresets for KDE (Phillips & Tai, 2020; 2018) fall under this category. These approaches aim to find a small set of possibly weighted samples for a specific objective function such that the result obtained by applying the function to this small set is within a small approximation error of the result obtained by applying the objective function on a complete dataset. The issue with these approaches is that of efficiency. Most of these algorithms require complicated computation over the entire data. Some streaming algorithms were recently proposed for coresets for KDE (Karnin & Liberty, 2019). However, even these algorithms need to perform $\mathcal{O}(m)$ ($m$ is compactor size) computationally expensive operations per sample for large chunks of size ($m$), making them unsuitable for our purposes. (2) **Dimensionality reduction:** These approaches aim to reduce the width of the data matrix. Approaches such as Principle Component Analysis (PCA) are computationally expensive and require iterative computation over the entire dataset. Random projections are an efficient streaming algorithm for dimensionality reduction. However, this approach leads to compressed data that increases linearly with the original data size. As we can see, existing approaches fall short of the requirements in our problem statement.

## 3 BACKGROUND

### 3.1 HISTOGRAMS AND KERNEL DENSITY ESTIMATION

Histograms and KDE (Scott, 2015; Scott & Sain, 2004) are popular methods to estimate the density of a distribution given a finite i.i.d. sample of $n$ points in $R^d$ drawn from the true density, say $f(x)$.

**Histogram:** Histogram divides the support $S \subset R^d$ of the data into multiple partitions. It then uses the counts in every partition to predict the density, $\hat{f}_H(x)$, at a point $x$. Formally the density predicted at the point $x \in S$ is given by

$$\hat{f}_H(x) = \frac{\mathcal{C}(\text{bin}(x))}{n\mathcal{V}(\text{bin}(x))}$$

where $\text{bin}(x)$ identifies the partition of $x$, $\mathcal{C}(b)$ and $\mathcal{V}(b)$ measures the the number of samples in partition $b$ and the volume of partition $b$ respectively. $\hat{f}_H(x)$ integrates to 1 and hence $\hat{f}_H(x)$ is also an estimate of the underlying density function $f(x)$. Regular histograms use hyper-cube partitions of width $B$ aligned with the data axes. As $B$ increases, the bias of the estimate increases, and its variance decreases. Histograms suffer from bin-edge problems where a slight change in data across the bin's edge can change predictions significantly.

**Kernel Density Estimation(KDE):** KDE provides a smoother estimate of $f(x)$ which resolves the bin-edge problem of histograms. For a positive semi-definite kernel function $k(x,y) : R^d \times R^d \to R$

and data, say $D$, the KDE at point $x$ is defined as

$$\hat{f}_K(x) = \text{KDE}(x) = \frac{1}{n}\sum_{i=1}^{n} k(x, x_i) \quad \text{where } x_i \in D$$

Kernel functions are positive, symmetric, and may be normalized to integrate to 1. Gaussian, Epanechnikov, Uniform, (Friedman et al., 2001) are some of the most widely used kernels. A smoothing parameter $B$ also parameterizes the kernel function and determines the standard deviation parameter for the Gaussian kernel function. For uniform and Epanechnikov kernel functions, $B$ is the window width around $x$ where the kernel is non-zero. As $B$ increases, the bias of KDE increases, and its variance decreases.

Histograms estimator and KDE both uniformly converge to underlying true distribution asymptotically. However, both suffer from the curse of dimensionality. To get a decent estimate of density in high dimensions, the number of samples needed is exponential in dimensions. For the density estimation task, dimensions of 4-50 are considered large. (Wang & Scott, 2019)

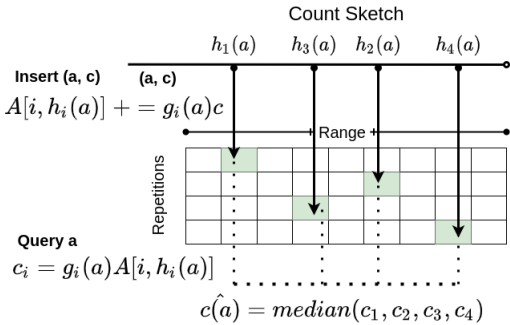

Figure 1: Countsketch, sketching, and query

### 3.2 COUNT SKETCHES

The count sketch (CS) (Cormode & Muthukrishnan, 2009; Charikar et al., 2002), along with its variants, is one of the most popular probabilistic data structures used for the heavy hitter problem. Given a stream of $(a_t, c_t)$ key-value pairs, $a_t \in \mathcal{U}$, CS stores the compressed total counts for each of the keys in a small $K \times R$ array of integers and can be queried to retrieve this total count, $\mathcal{C}(a_t)$. CS offers a probabilistic solution in memory logarithmic in a total number of unique keys. There is a standard memory accuracy trade-off for CS. Let $m$ be the number of distinct keys and $\mathbf{C}$ be the vector of counts indexed by each key. For count median sketch (Charikar et al., 2002), the $(\epsilon, \delta)$ guarantee $\mathbb{P}(|\hat{\mathcal{C}}(a) - \mathcal{C}(a)| > \epsilon ||\mathbf{C}||_2) \leq \delta$ is achieved using $\mathcal{O}(\frac{1}{\epsilon^2}\frac{1}{\delta}(\log m + \log|\mathcal{U}|))$ space. (Chakrabati, 2020). As seen from the above equation, the approximation accuracy for a particular key depends on how it compares to the $||\mathbf{C}||_2$. Specifically, CS can give an excellent approximation for keys with the highest values in a setting where most other keys have very low values. More discussion on CS can be found in Appendix F.

### 3.3 LOCALITY SENSITIVE HASHING

Locality-sensitive hashing(LSH)(Darrell et al., 2005) is a popular approach to solving approximate near-neighbor problems. If a function $h : \mathcal{U} \to \{0, ...r-1\}$ for some $r$, is randomly drawn from the LSH family $\mathcal{L}$, the probability of collision of the hash values for two distinct elements $a_1$ and $a_2$ is

$$\mathbb{P}_{h \in \mathcal{L}}(h(a_1) == h(a_2)) \propto \text{Sim}(a_1, a_2)$$

Where $\text{Sim}(a_1, a_2)$ is some similarity metric corresponding to the LSH family. The probability of collision is referred to as the kernel of the LSH family, generally denoted by $\phi(.,.)$. Most kernels are positive, bounded, symmetric, and reflective. We can use $p$ independent LSH functions, $h_1, h_2, ...h_p$ to obtain a LSH function, $h^{(p)}(a) = (h_1(a), h_2(a), ..., h_p(a))$. The function $h^{(p)}$ has kernel $\psi(.,.) = \phi(.,.)^p$. We call $p$ the power of the LSH function. Popular LSH functions for $\mathcal{U} = R^d$ are L2-LSH, L1-LSH and SRP (signed random projection). More details on LSH functions can be found in (Darrell et al., 2005)

### 3.4 UNIFORM SAMPLING FROM CONVEX POLYTOPES

Uniform sampling from convex spaces is a well-studied problem (Bélisle et al., 1993; Chen et al., 2017). For general convex polytopes, this is achieved by finding a point inside the polytope using convex feasibility algorithms and then running an MCMC walk inside the polytope to generate a point with uniform probability. In the case of regular convex polytopes like hypercubes and parallelopiped, uniform sampling is much simpler. Sampling a data point at random in a $d$-dimensional hypercube of width 1 is equivalent to uniformly sampling $d$ real values in the interval $[0, 1]$. For sampling within a $d$-dimensional parallelopiped, we first locate $(d-1)$-dimensional hyperplane parallel to each face at a distance drawn uniformly from $[0, B]$ where $B$ is the width of parallelopiped in that direction. The sampled point is the intersection of these $(d-1)$-dimensional hyperplanes.

Table 1: bin$(x)$ for different partitioning schemes

| Partitioning Scheme | Parameters | bin$(x) : R^d \rightarrow \mathbf{N}^d$ | Sampling $s \in R^d$ from $b \in \mathbf{N}^d$ |
|---|---|---|---|
| Regular histogram | $B \in R$ | bin$(x)_i = \lfloor x_i/B \rfloor$ | $r_i \sim U(0,1), r \in R^d$
$s = B(b+r)$ |
| Aligned histogram | $\mathbf{B} \in R^d$ | bin$(x)_i = \lfloor x_i/\mathbf{B}_i \rfloor$ | $r_i \sim U(0,1), r \in R^d$
$s = \mathbf{B} \circ (b+r)$ |
| L1/L2-LSH | $\mathbf{W} \in R^{d \times d}$
$\mathbf{B} \in R^d, t \in R^d$ | bin$(x)_i = (\lfloor \langle x, \mathbf{W}_i \rangle + t_i \rfloor / \mathbf{B}_i \rfloor)$ | $r_i \sim U(0,1), r \in R^d$
$y = \mathbf{B} \circ (b+r)$
$s = \text{solve}(\mathbf{W}s = y - t)$ |
| SRP | $\mathbf{W} \in R^{k \times d}$ | bin$(x)_i = (\text{sign}(\langle x, \mathbf{W}_i \rangle))$ | MCMC with constraints,
$\text{sign}(\langle x, \mathbf{W}_i \rangle) = b_i$
+ bounding box |

## 4 DENSITY SKETCHES

In DS, we aim to build a compressed non-parametric estimation object in an efficient streaming fashion. As KDEs give a better approximation of underlying function $f(x)$ than histograms, we want to build DS as a compressed KDE object. To achieve this, we use a nice connection between KDE and Histograms with an LSH-based partition function.

### 4.1 HISTOGRAM WITH LSH-BASED PARTITION AND KERNEL DENSITY ESTIMATES

Any LSH function on $R^d$ will partition the space into different bins. Specifically, if power $d$ L1/L2-LSH, these partitions will be polytopes in $R^d$. Similarly, power k SRP would give conical partitions with hyper-plane boundaries. We can employ a histogram-based estimation strategy on the top of these randomly drawn partitions. The density estimate using such a histogram would be

$$\hat{f}_H(x) \propto \frac{1}{n} \sum_{i=1}^{n} \mathcal{I}(x_i \in \text{bin}(x)) \quad \text{where } x_i \in D$$

where $\mathcal{I}$ is a indicator function. This estimate of the density has an expected value (over random partitions) equal to the KDE estimate, say $\hat{f}_\phi(x)$ with the corresponding LSH kernel, $\phi(.,.)$

$$\mathbf{E}_p(\hat{f}_H(x)) = \frac{1}{n} \sum_{i=1}^{n} P(x_i \in \text{bin}(x)) = \frac{1}{n} \sum_{i=1}^{n} \phi(x_i, x) = \hat{f}_\phi(x)$$

The expectation is over random partitions. This connection between randomized histograms and KDE was first observed in (Coleman & Shrivastava, 2020). To better approximate KDE, we can combine results from multiple histograms with independent LSH functions. For example, if we use $m$ independent histograms, say $H_1, H_2, ..., H_m$, then the density estimate can be written as

$$\hat{f}_H^{(m)}(x) = \frac{1}{m} \sum_{i=1}^{m} \hat{f}_{H_i}(x)$$

We can sample a data point from this set by first choosing a histogram randomly and then sampling a point from that histogram. One can check that the sampling distribution, thus obtained, is $\hat{f}_H^{(m)}(x)$.

### 4.2 CONSTRUCTING DENSITY SKETCHES

Now that we have reduced the problem of KDE approximation to histograms, we will now show how to obtain a compressed representation of a histogram in a streaming fashion. We also show how to generate samples from this representation. First, let us establish some notation we will use

**Notation:** (1) Data $D$ consists of $n$ i.i.d samples of dimension $d$ drawn from true distribution $f(x)$ : $S \subset R^d \rightarrow R$. (2) bin$(x)$: ID of the partition in which point $x$ falls. In the case of p-power LSH functions, bin$(x) : R^d \rightarrow N^p$, and each bin can be identified with a unique tuple of $p$ integers. In a regular histogram, we have a tuple of $d$ integers. For example, in regular histogram with width $B$, bin$(x)_i = \lfloor x_i/B \rfloor$. bin is generally parameterised with bandwidth parameter $B$ which measures the size of the partition. Some partitioning schemes and sampling algorithms are mentioned in Table 1 (3) A CS $\mathcal{M}$ with range $R$ and repetitions $K$ as described in section 3. (4) $\mathcal{H}$: Augmented min-heap of size $H$ used with $\mathcal{M}$. Hence for a given partitioning scheme (bin, $B$), DS is parameterized by $(K, R, H)$ and includes two data structures $\mathcal{M}(K, R)$ and $\mathcal{H}(H)$.

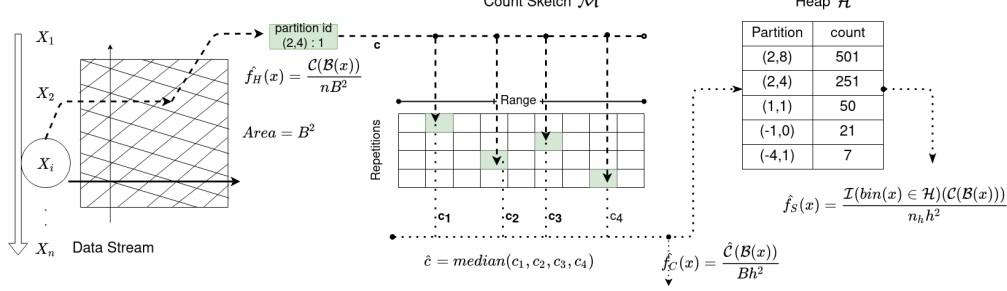

Figure 2: Overview of the sketching algorithm

The histogram has an exponential (in $d$) number of partitions. Hence, in high dimensions, it is impractical to store histograms. However, most high dimensional real data is clustered and thus has highly sparse histograms. This does not help with histograms, as post-pruning of histograms still requires us to build and enumerate them. Nevertheless, the sparsity in histogram makes it a good candidate for heavy hitter problems. We use CS, $\mathcal{M}$, to store a compressed version of the histogram. Unfortunately, sampling with just $\mathcal{M}$ does not have an efficient solution. We maintain a set of heavy partitions for sampling in the min-heap $\mathcal{H}$. We will discuss sampling in later subsections.

**sketching $\mathcal{M}$ :** As shown in figure 4 and algorithm 1, we process the data in a streaming fashion. For each data point, say $x$, we find the partition $b = \text{bin}(x)$. We increment the count of $b$ by 1 by inserting $(b, 1)$ into $\mathcal{M}$. Along with each insertion, we also update $\mathcal{H}$. If the $\mathcal{H}$ is not at its capacity, we insert this $b$ into the heap along with its updated count $\hat{\mathcal{C}}(b)$. If the heap is at its capacity, we check $b$'s updated count against the minimum of the $\mathcal{H}$. If $b$'s count is greater, we pop the minimum element from the heap and insert $(b, \hat{\mathcal{C}}(b))$.

### 4.3 $\hat{f}_C(x)$: ESTIMATE OF DENSITY AT A POINT

We can use $\mathcal{M}$ for querying the density estimate at a particular point. The algorithm for querying is presented in Algorithm 2 and is explained in figure 2. Reusing notation from 3.1, the density predicted by the histogram can be written as $\hat{f}_H(x)$. When using the sketch, instead of actual $\mathcal{C}(\text{bin}(x))$, we would use its estimate from $\mathcal{M}$. Let this estimate be $\hat{\mathcal{C}}(\text{bin}(x))$. Then we can write the density predicted using count-sketch as $\hat{f}_C(x)$

$$\hat{f}_H(x) = \frac{\mathcal{C}(\text{bin}(x))}{n\mathcal{V}(\text{bin}(x))} \qquad \hat{f}_C(x) = \frac{\hat{\mathcal{C}}(\text{bin}(x))}{n\mathcal{V}(\text{bin}(x))}$$

We know from CS literature that $\hat{\mathcal{C}}(\text{bin}(x))$ is closely distributed around $\mathcal{C}(\text{bin}(x))$ and so we can expect $\hat{f}_C(x)$ to be close to $\hat{f}_H(x)$ and hence to $f(x)$. Note that though $\hat{f}_C(x)$ is a good estimate of density at a point x, the function $\hat{f}_C(.)$ is not a density function as it does not integrate to 1.

### 4.4 $\hat{f}_C^*(x)$: ESTIMATE OF DENSITY FUNCTION

To obtain a density function from the sketches, we have to normalize the function $\hat{f}_C(x)$ over the support. We can write $\hat{f}_C^*(x)$ as

$$\hat{f}_C^*(x) \propto \hat{\mathcal{C}}(x) \qquad \hat{f}_C^*(x) = \frac{\hat{\mathcal{C}}(x))}{\int \hat{\mathcal{C}}(x)dx} = \frac{\hat{\mathcal{C}}(x)}{\mathcal{V}(\text{bin}(x))\sum_{b\in\text{bins}(S)}\hat{\mathcal{C}}(b)} = \frac{\hat{\mathcal{C}}(x)}{\mathcal{V}(\text{bin}(x))\hat{n}}$$

It is easy to check the integral can be written as the sum over all the bins in the support. As is clear from the equations for $\hat{f}_C^*(x)$ and $\hat{f}_C(x)$, $n = \sum_{b\in\text{bins}(S)}\mathcal{C}(b)$ , is replaced by $\hat{n} = \sum_{b\in\text{bins}(S)}\hat{\mathcal{C}}(b)$ to get a density function. We can check that $\hat{n}$ is an estimate of $n$ using an estimate of count for each bin from the DS.

### 4.5 $\hat{f}_S(x)$: SAMPLING FROM DENSITY SKETCHES

$\mathcal{M}$ is a good enough representation for querying the density at a point. However, it is not the best data structure to generate samples efficiently. One naive way of sampling from these sketches is to

**Algorithm 1: Constructing density sketch of $f(x)$**

**Result:** Density Sketch (DS)
$f(x) : R^d \to R$ : true distribution
$x_1, \ldots x_n \sim f(x)$ : sample drawn from $f(x)$
$\text{bin}(x) : S \to N^d$: partition function
$\mathcal{M}$ : CS with range R, repetitions K
$\mathcal{H}(H)$ : min-heap to store top H partitions

**for** $i \leftarrow 1$ **to** $n$ **do**
    $b = \text{bin} x_i$
    $\mathcal{M}.insert(b, 1)$
    $c = \mathcal{M}.query(b)$
    $\mathcal{H}.update(b, c)$

**Algorithm 2: query $\hat{f}_C(y), y \in R^d$**

**Result:** $\hat{f}_C(y)$
$y \in R^d$
$b = \text{bin} y$
$c = \mathcal{M}.query(b)$
return $(c/(n\mathcal{V}(b)))$

**Algorithm 3: sample $y \in R^d$ such $y \sim \hat{f}_S(x)$**

**Result:** y: sample from $f_S(y)$
P : categorical distribution over bins s.t.
  –    $P(b) = (\mathcal{H}[b]/n_h)$ if $b \in \mathcal{H}$
  –    $P(b) = 0$ if $b \notin \mathcal{H}$
$b \sim P$
$y = \text{UniformRandomPoint}(b)$
return $y$

randomly select a point in support of $f(x)$ and then do a rejection sampling using estimate $\hat{f}_C(x)$. However, given the enormous volume of support in high dimensions, this method is bound to be immensely inefficient. Another way is to choose a partition with probability proportional to the count of elements in that partition and then sample a random point from this chosen partition. It is easy to check that the probability of sampling a point $x$ in this manner, precisely, is $\hat{f}_H(x)$ if we use exact counts and $\hat{f}_C^*(x)$ if we use approximate counts from CS. However, given that number of bins is exponential in dimension, sampling a bin proportional to its counts requires prohibitive memory and computation. This is why we needed a CS in the first place. Here, we further approximate the distribution by storing only top $H$ partitions which contain most data points and discarding other partitions. As mentioned in 1, we can efficiently maintain top $H$ partitions with an augmented heap $\mathcal{H}$. We then sample a partition present in this heap with probability proportional to its count and sample a random data point from this partition (Algorithm 3). The probability of sampling a data point whose bin is not present augmented heap is then zero. The distribution of this sampling algorithm is,

$$\hat{f}_S(x) = \mathcal{I}(\text{bin}(x) \in \mathcal{H}) \frac{\hat{\mathcal{C}}(\text{bin}(x))}{\hat{n}_h \mathcal{V}(\text{bin}(x))}$$

where $\hat{n}_h = \sum_{b \in \mathcal{H}} \hat{\mathcal{C}}(b)$ is the count-sketch estimate of the total number of elements captured in all partitions present in the heap. $\mathcal{I}(.)$ is the indicator function with values 0 or 1 evaluating the boolean statement inside it. Let $\rho_h = \hat{n}_h/\hat{n}$ be the capture ratio of heap. It is easy to see that as the capture ratio tends to 1, $\hat{f}_S(x)$ tends to $\hat{f}_C^*(x)$. Note that $\hat{f}_S(x)$ is a density function.

## 5 ANALYSIS

Histogram and Kernel Density Estimators are well-studied non-parametric estimators of density. Both of these estimators are shown to be capable of approximating a large class of functions (Scott, 2015). For example, with the condition of Lipschitz Continuity on $f(x)$, we can prove that pointwise MSE($\hat{f}_H(x)$ converges to 0 at a rate of $\mathcal{O}(n^{-2/3})$. Better results can be obtained for functions that have continuous derivatives. In our analysis, we make assumptions along those made in (Scott, 2015); specifically, the existence and boundedness of all function-dependent terms that appear in the theorems below. We refer the reader to (Scott, 2015) for an in-depth discussion on assumptions.

We restrict our analysis to convergence in probability for all the estimators discussed in this paper, which is the standard (Scott, 2015). In this section, we consider the regular histogram partitioning scheme and show that our sampling distribution $\hat{f}_S(x)$ is an approximation of underlying distribution $f(x)$ and converge to it. However, a similar analysis holds even for random partitioning schemes / KDE and is skipped here.

**Mean integrated square error(MISE)**: MISE of an estimator of function is a widely used tool to analyze the performance of a density estimator.

$$\text{MISE}(\hat{f}) = \mathbf{E}\left[\int (\hat{f}(x) - f(x))^2 dx\right]$$

A density estimator, with MISE asymptotically tending to zero, is a consistent estimator of true density and converges to it in probability. We would use this tool to make statements about the convergence of our estimators. By Fubini's theorem, MISE is equal to IMSE (Integrated mean square error).

$$\text{MISE}(\hat{f}) = \text{IMSE}(\hat{f}) = \int \mathbf{E}\left[(\hat{f}(x) - f(x))^2)\right] dx$$

We now present our main result of the paper,

**Theorem 1** (**Main Theorem: $\hat{\mathbf{f}}_{\mathbf{S}}(\mathbf{x})$ to $\mathbf{f}(\mathbf{x})$** ). *The probability density function of sampling, $\hat{f}_S(x)$, using a DS over regular histogram of width $B$, with parameters(K,R,H) created with $n$ i.i.d samples from original density function $f(x)$, has an IMSE given by*

$$IMSE(\hat{f}_S(x)) \leq 12(1-\rho_h)^2 + 3(1+2\epsilon)\left(\frac{1}{nB^d} + \frac{\mathcal{G}(f)}{n} + o\left(\frac{1}{n}\right) + \frac{n_{nzp}-1}{KRnB^d}\right)$$

$$+ 3(1+3\epsilon)\left(\frac{B^2 d}{4}\mathcal{G}(\|\nabla f\|_2)\right) + 3\epsilon\left(1 + 2\mathcal{G}(f) + B\sqrt{d}\int_{x \in S}(f(x)\|\nabla f\|_2)\right)$$

*with probability $(1-\delta)$ , where $\delta = \frac{n_{nzp}}{\epsilon^2 nKR}$, $n_{nzp}$ is the number of non-empty bins in histogram, $\rho_h$ is the estimated capture ratio as described in section 4.5 and $\mathcal{G}(g)$ is roughness defined as $\int g(x)^2 dx$*

The dependence of IMSE on properties of $f(x)$, such as roughness, is standard (Scott, 2015) and cannot be avoided.

**Interpretation** The estimator $\hat{f}_S(x)$ of $f(x)$ is obtained by a series of approximations from $f(x) \to \hat{f}_H(x) \to \hat{f}_C(x) \to \hat{f}_C^*(x) \to \hat{f}_S(x)$. Hence to interpret this result, we break down the result above into multiple theorems enabling the reader to easily notice which step of approximations leads to what terms in the theorem above. We provide these details in Appendix. We notice a few things from the theorem below

- Similar to the standard analysis for histograms, the curse of dimensionality also manifests in our theorem. $B$ should go to zero and $n$ should increase faster than the rate at which $B^d/n_{nzp}$ decreases (condition 1). As compared to standard histograms, this requires $n$ to grow faster. With these conditions on $B$ and $n$, it is clear how the second and third terms go to zero.
- The magnitude of the fourth term is controlled via $\epsilon$. The above statement is true for any $\delta$ and $\epsilon$ that are related via the expression $\delta = \left(n_{nzp}/(\epsilon^2 nKR)\right)$. Choose arbitrarily small $\epsilon$ and $\delta$, and we can achieve it with large enough $n/n_{nzp}$ or by providing more intermediate resources and making $KR$ large enough. For a fixed resource $KR$, this term goes to zero asymptotically with $n$ growing faster than $n_{nzp}$, which is a sub-condition of condition 1.
- The term $12(1-\rho_h)^2$ shows the effect of truncation that occurs due to using only heavy partitions. As can be seen, this term is data dependent, and IMSE does not depend directly on $H$ (number of partitions) but $\rho_h$. Suppose we can capture the entire data in the heap (i.e., setting $H=n_{nzp}$), then the term adds no penalty to IMSE. $H$, via $\rho_h$ controls the accuracy-memory trade-off of DS.

## 6 DISCUSSION

**curse of dimensionality:** As DS are built over Histograms, they inherit the curse from Histograms: i.e., the number of samples needed increases exponentially with dimension. With increased data collection, the issue of the unavailability of large amounts of data is fast vanishing. We want to emphasize that DS' advantages are best seen when data is humongous. DS can absorb tons of data and give better density estimates and samples without increasing memory usage. Also, most real data in high dimensions is clustered or stays on a low-dimensional manifold. DS, throw away empty bins, and only store the histogram's populated bins. DS can deal with the curse of dimensionality better than Histograms. **DS on original data space:** Some data types, like images, do not reside in a space where the usual distances or cosine similarities imply conceptual similarity. On these data types, DS will not perform well. One way is to learn a transformation and create sketches of the transformed data. While this will give better performance in practice, we might lose theoretical guarantees for certain transformations.

## 7 EXPERIMENTS

**Visualization of samples from density sketches:** In the first set of experiments, we provide a sanity check for DS in the form of visualization of data generated from DS. (1) In the first experiment,

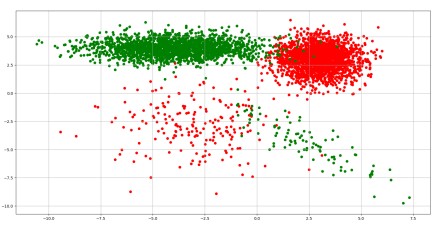

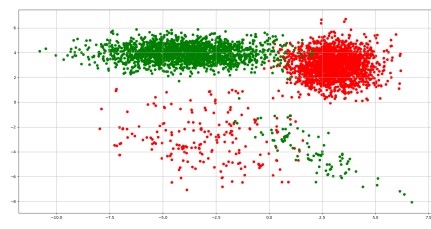

(a) sample drawn from true distribution

(b) sample drawn from Density Sketch

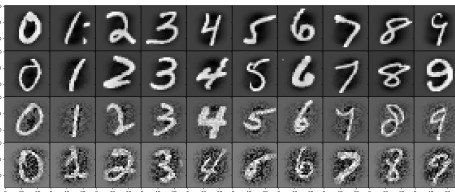

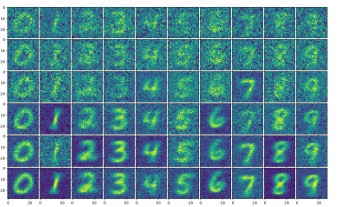

(c) MNIST Sample from Density Sketch using L2-LSH partitioning width: 0.01,0.1,1,10 from top to bottom

(d) MNIST Sample from Density Sketch using conical partitions (multiple signed random projections (SRP)). The rows have varying number of SRPs used - increasing from top to bottom

Figure 4: Visualization of samples drawn from Density Sketch. (a-b) DS captures density information. (c) Higher power LSH functions lead to fine DS behaving like random sample (d) Coarser partitions lead to samples that resemble "avg" of samples in data

figure 4 (a) shows the samples drawn from the actual multi-gaussian distribution, whereas figure 4 (b) shows the samples drawn from the DS built on the samples from the true distribution. In this experiment, we use L2-LSH partitions with a B=0.25, K=3, R=50000, and H=3000. As can be seen, the two samples are indistinguishable. So DS does capture the density information. (2) Figure 4 (c) shows some samples drawn from DS built over the MNIST dataset (Chang & Lin, 2011) with varying bin-width sizes. (again R=50000, K=3 and H=5000) . We should notice that MNIST with 784 dimensions and 60K samples is not an ideal dataset for DS. In fact, with L2-LSH partitions the data would be so scattered that every sample is contained in its bin. If we make the bin-width finer, we should sample data points very close to the random sample from the original data. So in the worst conditions, DS converges to a random sample which we know is a good representation of data. 4 (d) shows results again with MNIST. However, here we have created conical partitions (created using multiple signed random projections). While L2-LSH partitions use power 784 L2-LSH functions, in this experiment, we use a smaller number of SRP functions(10-25, increases as we go to lower rows in the image), thus promoting clustering. As expected, this coarse partitioning does show significant clustering; hence, the images drawn from the partitions look like the average of multiple samples from the original data. The results support that DS can create samples that resemble the original data.

**Evaluation of Samples on Classification Tasks:** For most datasets, it is not possible to inspect samples visually. Hence we evaluate the quality of samples from DS by using them to train classification models. In these experiments, the data loader of the training algorithm is replaced with a sampler from DS. This sampler returns a training batch when requested by the algorithm. All the experiments are performed on Tesla P-100 GPUs with 16GB memory.

**Datasets:** We choose big datasets from the liblinear website (Chang & Lin, 2011), which satisfy the constraints of 1) data dimension less than 100 and 2) the number of samples per class greater than 1,000,000. Large Datasets is the main application domain for DS. Thus, we have datasets of Higgs (10M samples, 28 dimensions) and Susy (5M samples, 18 dimensions) for our experiments.

**Baselines:** For baselines, we consider random samples of the same size and Liberty coresets proposed by (Karnin & Liberty, 2019) to compare DS performance. For Liberty coresets, we use $m = 100$ as for larger m the process is very slow. Dimensionality reduction via random projections is another streaming algorithm. Still, in these datasets, we cannot get significant compression using

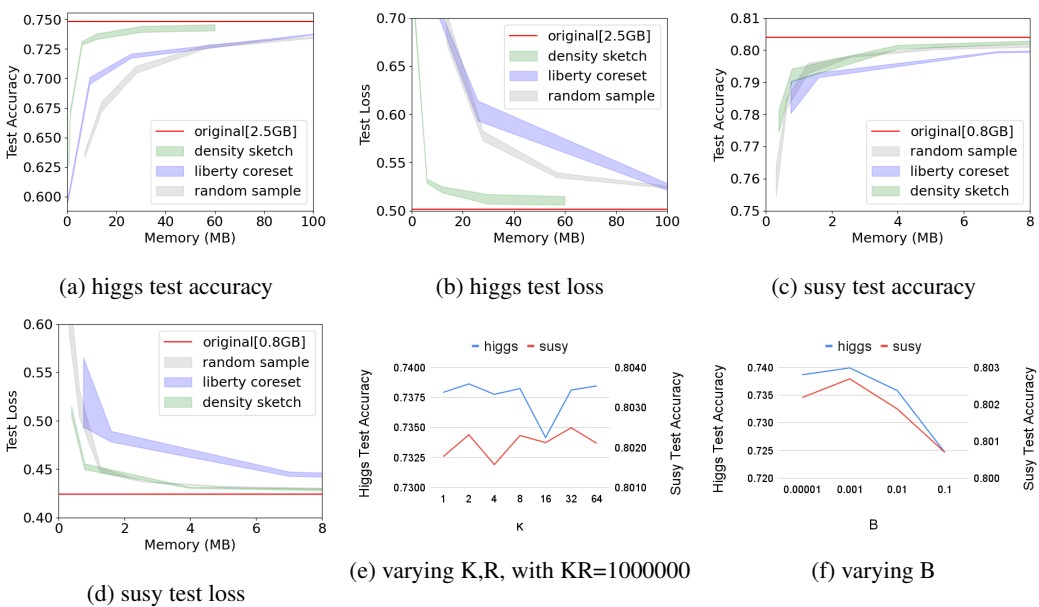

Figure 5: (a-d) Performance of density sketches, Liberty coresets and random sample on downstream classification task. The bar widths refer to $2 \times std$. DS performs consistently better at all memory sizes (e) Performance of DS is stable with K,R configs for decent budget KR=1000000, (f) Optimal B exists for DS performance. larger and smaller values of B lead to performance degradation

dimensionality reduction (it can be $d\times$ at best, where $d$ is the dimension). So we cannot compare against this method. For a more detailed discussion on baselines, refer to appendix D.

**Results:** The DS has parameters: partition function(bandwidth B). We use L2-LSH partitions, sketch parameters K, R, and heap parameter H in all our experiments. The memory of the DS used for sampling is affected by only the heap parameter (see appendix F for details on memory computation). In figure 5(f), we use the config (K=4,R=250000,H=100000) and vary B. It is clear from the figure that B=0.001 works best for these datasets. Lower and higher values of B affect the performance adversely. Larger B implies that we will capture more space than needed in a single partition, and smaller B implies that we will capture lesser data in the heap. So it is expected that a sweet spot for B exists. In figure 5(e), we fix (B=0.01,H=100000, KR=1000000) and vary K from 1 to 64. We can see that for a reasonable memory budget the results are stable with varying K. For the experiments in figure 5(a-d) we fix (B=0.01, K=5, R=250000) and vary H. This gives us DS of different sizes. We plot test accuracy and losses for DS, random samples and Liberty coresets for different sizes of memory used. The width of the band signifies the $2 \times$ std-dev of performance on three independent runs. As can be seen, for the "Higgs" dataset, the model's accuracy achieved on original data of size 2.5GB can be closely reached by using a DS of size 50MB. So we get around **50x compression** ! We see similar results for datasets of "Susy" (**100x compression**, 0.8GB) as well. The results show that DS is much more informative than Random Sample and Liberty Coresets. For more details on running the experiments (data processing, memory measurements, etc.), refer to Appendix F.

**Estimation of statistical properties of dataset:** We also perform the experiments on the covariance estimation task. The observations are similar to the classification experiment. DS performs better than the corresponding random sample. The results are presented in Appendix F for the shortage of space.

## 8 CONCLUSION

In this paper, we talk about Density Sketches, a streaming algorithm to construct a summary of density distribution from data. We show that new samples generated from this sketch asymptotically converge to the underlying distribution. Thus, DS comes with theoretical guarantees. Additionally, the cheap nature of online updates in Density Sketches, makes it an attractive alternative to constructing coresets for the data. In terms of coresets, DS can be viewed as a compressed form of randomized grid-coresets - one of the oldest forms of coresets.

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

Table 2

| Notation | Description |
|---|---|
| $f(x)$ | true density distribution |
| $\hat{f}_H(x)$ | histogram estimate of f(x) |
| $\text{bin}(x)$ | partition id of point $x$ |
| $\text{bins}(S)$ | enumerates all bins in the support S |
| $\mathcal{C}(b)$ | count of elements in the bin id $b$ |
| $\mathcal{C}(x)$ | count of elements in the bin$(x)$ |
| $\hat{\mathcal{C}}(b)$ | estimate count of elements in the bin id $b$ |
| $\mathcal{V}(b)$ | volume of bin id $b$ |
| $D$ | data |
| $k(,)$ | kernel function : $R \times R \to R$ |
| $B$ | $B \in R$, scalar bin width |
| $\mathbf{B}$ | $\mathbf{B} \in R^d$ vector of bin widths in different directions |
| $h(x)$ | $\mathcal{U}\{0, ...R-1\}$ hash function |
| $g(x)$ | $\mathcal{U} \to \{-1, +1\}$ hash function |
| $W \in R^{k \times d}$ | weight matrix for SRP / L1-L2 LSH |
| $\mathcal{I}$ | indicator function |
| $\mathbf{E}$ | Expected value |
| $\hat{f}_\phi(x)$ | KDE estimate with kernel $\phi$ |
| $\mathcal{M}$ | Count sketch |
| $K, R$ | parameters of count sketch |
| $n$ | total number elements |
| $\hat{n}$ | CS estimate of total number of elements |
| $n_h$ | number of elements in the heap |
| $\hat{n}_h$ | CS estimate of number of elements in the heap |
| $\rho_h$ | capture ratio defined by $\hat{n}_h/\hat{n}$ |

# A APPENDIX

## A.1 NOTATION

## A.2 THEOREM 2 : $f_H$ TO $f$

While estimating true distribution $f(x) : R^d \to R$, the integrated mean square error (IMSE) for the estimator $\hat{f}_H(x)$ using regular histogram with width h and number of samples n, is

$$\text{IMSE}(\hat{f}_H) \leq \frac{1}{nh^d} + \frac{\mathcal{G}(f)}{n} + o\left(\frac{1}{n}\right) + \frac{h^2 d}{4}\mathcal{G}(\|\nabla f\|_2)$$

Specifically, we have integrated variance (IV) and integrated square bias (ISB) as follows

$$\text{IV}(\hat{f}_H) = \frac{1}{nh^d} + \frac{\mathcal{G}(f)}{n} + o\left(\frac{1}{n}\right)$$

and

$$\text{ISB}(\hat{f}_H) \leq \frac{h^2 d}{4}\mathcal{G}(\|\nabla f\|_2)$$

where $\mathcal{G}(\phi)$ is the roughness of the function $\phi$ defined as $\mathcal{G}(\phi) = \int \phi^2(x)dx$

*Proof.* Let $x \in S$ where S is the support of the distribution. Let bin$(x)$ determine the bin of point $x$, bins$(S)$ enumerate all the bins that lie inside the support $S$ of the distribution $f(x)$ . Let $V(\text{bin}(x))$ is volume of bin in which $x$ lies. Equivalently, we can also use $V(b)$ to denote volume of bin $b$. For standard histogram, $V(b) = h^d$

$$\hat{f}_H(x) = \frac{1}{nV(\text{bin}(x))} \sum_{i=1}^{n} \mathcal{I}(x_i \in \text{bin}(x)) \tag{1}$$

First let us consider the integrated variance.

$$\text{IV} = \int_{x \in S} \text{Var}(\hat{f}_H(x)) dx = \sum_{b \in \text{bins}(S)} \int_{x \in b} \text{Var}(\hat{f}_H(x)) dx \tag{2}$$

For a particular bin $b$, the variance is constant at all values of $x$ inside it. Also for a particular $x$ in bin $b$, we can write the following for $\text{Var}(\hat{f}_H(x))$ using independence of samples.

$$\text{Var}(\hat{f}_H(x)) = \frac{1}{nV(\text{bin}(x))^2} \text{Var}(\mathcal{I}(x_i \in \text{bin}(x))) \tag{3}$$

Also $\text{Var}(\mathcal{I}(x_i \in b)) = p_b(1 - p_b)$ where $p_b$ is the probability of $x_i$ lying in bin $b$. That is, $p_b = \int_{x \in b} f(x) dx$

Using this in equation 2

$$\text{IV} = \sum_{b \in \text{bins}(S)} V(b) \frac{1}{nV^2(b)} p_b(1 - p_b) \tag{4}$$

Simplifying,

$$\text{IV} = \sum_{b \in \text{bins}(S)} \frac{1}{nV(b)} p_b(1 - p_b) \tag{5}$$

For standard histogram $V(b)$ is same across bins,

$$\text{IV} = \frac{1}{nV(b)} \left( \sum_{b \in \text{bins}(S)} p_b - \sum_{b \in \text{bins}(S)} p_b^2 \right) \tag{6}$$

$$= \frac{1}{nV(b)} \left( 1 - \sum_{b \in \text{bins}(S)} p_b^2 \right) \tag{7}$$

Using mean value theorem, we can write, $p_b = V(b) f(\xi_b)$ for some point $\xi_b \in b$.

$$\sum_{b \in bins} p_b^2 = \sum_{b \in bins} V(b)^2 f(\xi_b)^2 = V(b) \sum_{b \in bins} V(b) f(\xi_b)^2 \tag{8}$$

Using Riemann Integral approximation , we can write the following as the bin size reduces,

$$\sum_{b \in bins} V(b) f(\xi_b)^2 = \int_{x \in S} f^2(x) dx + o(1) \tag{9}$$

$\int_{x \in S} f^2(x) dx$ is also known as the roughness of the function. Let us denote it using $\mathcal{G}(f)$. Hence

$$\text{IV} = \frac{1}{nV(b)} (1 - V(b) (\mathcal{G}(f) + o(1))) \tag{10}$$

$$\text{IV} = \frac{1}{nV(b)} - \frac{\mathcal{G}(f)}{n} - o\left(\frac{1}{n}\right) \tag{11}$$

Putting $V(b) = h^d$

$$\text{IV} = \frac{1}{nh^d} - \frac{\mathcal{G}(f)}{n} - o\left(\frac{1}{n}\right) \tag{12}$$

Keeping only the leading term in the above expression,

$$\text{IV} = \mathcal{O}\left(\frac{1}{nh^d}\right) \tag{13}$$

Now let us look at the ISB for this estimator, $\text{ISB}(\hat{f}_H(x))$

$$\text{ISB}(\hat{f}_H(x)) = \int_{x \in S} (\mathbb{E}(\hat{f}_H(x) - f(x)))^2 dx \tag{14}$$

Let us look at the expected value of the estimator,

$$\mathbb{E}(\hat{f}_H(x)) = \frac{1}{V(\text{bin}(x))} \int_{t \in \text{bin}(x)} f(t)dt \tag{15}$$

Recall that $x \in R^d$. Using 2nd order multivariate taylor series expansion of this $f(t)$ around $x$, we get,

$$f(t) = f(x) + \langle t - x, \nabla f(x) \rangle + \frac{1}{2}(t-x)^\top \mathcal{H}(f(x))(t-x) \tag{16}$$

Here $\mathcal{H}(f(t))$ is the hessian of $f$ at $t$. Without the loss of generality let us look at the $\text{bin}(x) = [0, h]^d$ that is the bin at the origin. Let us say it is $b_0$

$$\int_{t \in b_0} f(t)dt = h^d f(x) + h^d \langle (\frac{h}{2} - x, \nabla f(x) \rangle + O(h^{d+2}) \tag{17}$$

Using eq 17 in eq 15, we get

$$\mathbb{E}(\hat{f}_H(x)) = f(x) + \langle (\frac{h}{2} - x), \nabla f(x) \rangle + O(h^2) \tag{18}$$

Hence, just keeping the leading term , we have

$$\text{Bias}(\hat{f}_H(x)) = \langle (\frac{h}{2} - x), \nabla f(x) \rangle \tag{19}$$

Now,

$$\int_{x \in b_0} \text{Bias}(\hat{f}_H(x))^2 dx = \int_{x \in b_0} \left( \langle \left( \frac{h}{2} - x \right), \nabla f(x) \rangle \right)^2 dx \tag{20}$$

Using Cauchy-Schwarz inequality, we get

$$\int_{x \in b_0} \text{Bias}(\hat{f}_H(x))^2 dx \leq \int_{x \in b_0} \|(\frac{h}{2} - x)\|_2^2 \|\nabla f(x)\|_2^2 dx \tag{21}$$

As $[h/2, h/2, ...h/2]$ is a mid point of the bin. The max norm of $x - h/2$ can be $h\sqrt{d}/2$

$$\int_{x \in b_0} \text{Bias}(\hat{f}_H(x))^2 dx \leq \frac{h^2 d}{4} \int_{x \in b_0} \|\nabla f(x)\|_2^2 dx \tag{22}$$

Now looking at ISB

$$\text{ISB}(\hat{f}_H) = \sum_{b \in bins} \int_{x \in b_0} \text{Bias}(\hat{f}_H(x))^2 dx \leq \frac{h^2 d}{4} \int_{x \in S} \|\nabla f(x)\|_2^2 dx \tag{23}$$

$$\text{ISB}(\hat{f}_H) \leq \frac{h^2 d}{4} \mathcal{G}(\|\nabla f\|_2) \tag{24}$$

## A.3 THEOREM 3: $f_C$ TO $f_H$

While estimating true distribution $f(x) : R^d \to R$, the integrated mean square error (IMSE) for the estimator $\hat{f}_C(x)$ using regular histogram with width $h$, number of samples $n$, and countsketch with range R, repetitions K and mean recovery, is

$$\text{IMSE}(\hat{f}_C) = \text{IMSE}(\hat{f}_H) + \frac{n_{nzp}}{KRnh^d}$$

where $n_{nzp}$ is the number of non-zero partitions. Specifically, we have

$$\text{IV}(\hat{f}_C) = \text{IV}(\hat{f}_H) + \frac{n_{nzp} - 1}{KRnh^d}$$

and

$$\text{ISB}(\hat{f}_C) = \text{ISB}(\hat{f}_H)$$

where $n_{nzp}$ is the number of non-zero bins/partitions. □

*Proof.* Consider a Countsketch with range $R$ and just one repetition (i.e. $K = 1$). Let it be parameterized by the randomly drawn hash functions $g : \text{bins}(S) \longrightarrow \{0, 1, 2, ..., R - 1\}$ and $s : \text{bins}(S) \longrightarrow \{-1, +1\}$. Let $\mathcal{C}(\text{bin}(x)) \sum_{i=1}^{n}(\mathcal{I}(x_i \in \text{bin}(x))$ is the count of elements that lie inside the $\text{bin}(x)$

The estimate of density at point $x$ can then be written as

$$\hat{f}_C(x) = \frac{1}{nV(\text{bin}(x))} \left( \mathcal{C}(\text{bin}(x)) + \sum_{i=1}^{n} \mathcal{I}\Big(x_i \notin \text{bin}(x) \wedge g(\text{bin}(x_i)) = g(\text{bin}(x))\Big) s(\text{bin}(x_i)) s(\text{bin}(x)) \right) \tag{25}$$

We can rewrite this as ,

$$\hat{f}_C(x) = \hat{f}_H(x) + \frac{1}{nV(\text{bin}(x))} \left( \sum_{i=1}^{n} \mathcal{I}\Big(x_i \notin \text{bin}(x) \wedge g(\text{bin}(x_i)) = g(\text{bin}(x))\Big) s(\text{bin}(x_i)) s(\text{bin}(x)) \right) \tag{26}$$

As $\mathbb{E}(s(b)) = 0$, it can be clearly seen that.

$$\mathbb{E}(\hat{f}_C(x)) = \mathbb{E}(\hat{f}_H(x)) \tag{27}$$

Hence, it follows that

$$\text{ISB}(\hat{f}_C(x)) = \text{ISB}(\hat{f}_H(x)) \tag{28}$$

It can be checked that each of the terms in the summation for right hand side of equation 26 including the terms in $\hat{f}_H(x)$ are independent to each other . i.e. covariance between them is zero. Hence we can write the variance of our estimator as,

$$\text{Var}(\hat{f}_C(x)) = \text{Var}(\hat{f}_H(x)) + \tag{29}$$

$$\frac{1}{nV^2(\text{bin}(x))} \text{Var}\left(\mathcal{I}\Big(x_i \notin \text{bin}(x) \wedge g(\text{bin}(x_i)) = g(\text{bin}(x))\Big) s(\text{bin}(x_i)) s(\text{bin}(x))\right) \tag{30}$$

$$\text{Var}(\hat{f}_C(x)) = \text{Var}(\hat{f}_H(x)) + \tag{31}$$

$$\frac{1}{nV^2(\text{bin}(x))} \mathbb{E}\left(\mathcal{I}\Big(x_i \notin \text{bin}(x) \wedge g(\text{bin}(x_i)) = g(\text{bin}(x))\Big)^2\right) \tag{32}$$

As square of indicator is just the indicator,

$$\text{Var}(\hat{f}_C(x)) = \text{Var}(\hat{f}_H(x)) + \tag{33}$$

$$\frac{1}{nV^2(\text{bin}(x))} \mathbb{E}\left(\mathcal{I}\Big(x_i \notin \text{bin}(x) \wedge g(\text{bin}(x_i)) = g(\text{bin}(x))\Big)\right) \tag{34}$$

$$Var(\hat{f}_C(x)) = Var(\hat{f}_H(x)) + \frac{1}{nV^2(\text{bin}(x))}(1 - p_{\text{bin}(x)})\frac{1}{R}) \tag{35}$$

Hence, IV is

$$\text{IV}(\hat{f}_C(x)) = \text{IV}(\hat{f}_H(x)) + \int_{x \in S} \frac{1}{nV^2(\text{bin}(x))}(1 - p_{\text{bin}(x)})\frac{1}{R}) \tag{36}$$

$$\text{IV}(\hat{f}_C(x)) = \text{IV}(\hat{f}_H(x)) + \sum_{b \in \text{bins}(S)} \int_{x \in b} \frac{1}{nV^2(b)}(1 - p_b)\frac{1}{R}) \tag{37}$$

$$\text{IV}(\hat{f}_C(x)) = \text{IV}(\hat{f}_H(x)) + \sum_{b \in \text{bins}(S)} \frac{1}{nV(b)}(1 - p_b)\frac{1}{R}) \tag{38}$$

Assuming standard partitions. $V(b) = h^d$ for all b

$$\text{IV}(\hat{f}_C(x)) = \text{IV}(\hat{f}_H(x)) + \frac{1}{nh^d}\frac{(n_{nzp}-1)}{R} \tag{39}$$

With mean recovery, with K repetitions, the analysis can be easily extended to get IV as

$$\text{IV}(\hat{f}_C(x)) = \text{IV}(\hat{f}_H(x)) + \frac{1}{nh^d}\frac{(n_{nzp}-1)}{KR} \tag{40}$$

The ISB remains same in this case. $\qquad\square$

## A.4 THEOREM 4: $\hat{f}_C^*$ TO $\hat{f}_C$

While estimating true distribution $f(x) : R^d \to R$, the IMSE for the estimator $\hat{f}_C^*(x)$ using regular histogram with width $h$ and number of samples $n$ and countsketch with parameters ($R$:range, $K$:repetitions), is related to the estimator $\hat{f}_C(x)$ as follows

$$\text{IMSE}(\hat{f}_C(x)) - \epsilon(N + 2M) \le \text{IMSE}(\hat{f}_C^*(x)) \le \text{IMSE}(\hat{f}_C(x)) + \epsilon(N + 2M)$$

Specifically,

$$\text{IV}(\hat{f}_C(x)) - 2\epsilon M \le \text{IV}(\hat{f}_C^*(x)) \le \text{IV}(\hat{f}_C(x)) + 2\epsilon M$$

and

$$\text{ISB}(\hat{f}_C(x)) - \epsilon N \le \text{ISB}(\hat{f}_C^*(x)) \le \text{ISB}(\hat{f}_C(x)) + \epsilon N$$

where

$$M \le \text{IV}(\hat{f}_C(x)) + 2(\mathcal{G}(f) + \frac{h^2 d}{4}\mathcal{G}(\|\nabla f\|_2) + h\sqrt{d}\int_{x\in S}(f(x)\|\nabla f\|_2))$$

$$N = (1 + \text{ISB}(\hat{f}_C(x)))$$

with probability $(1 - \delta)$ where $\delta = \frac{n_{nzp}}{\epsilon^2 nKR}$

*Proof.* Let us look at the estimator

$$\hat{f}_C^*(x) = \frac{\hat{\mathcal{C}}(\text{bin}(x))}{\mathcal{V}(\text{bin}(x))\sum_b \hat{\mathcal{C}}(b)} = \hat{f}_C(x) * \frac{n}{\hat{n}} \tag{41}$$

where $\hat{n} = \sum_b \hat{\mathcal{C}}(b)$ and $n = \sum_b \mathcal{C}(b)$ $\qquad\square$

**ñ and its relation to n:** Let us first analyse $\hat{n}$ and how it is related to n.

$$\hat{n} = \sum_{b\in\text{bins}(S)}\hat{\mathcal{C}}(b) = \sum_{b\in\text{bins}(S)}\sum_{i=1}^n\left(\mathcal{I}(x_i \in b) + \left(\mathcal{I}(x_i \notin b \wedge g(bin(x_i)) == g(b))s(bin(x_i))s(b)\right)\right) \tag{42}$$

$$\hat{n} = \sum_{b,i}\mathcal{I}(x_i \in b) + \mathcal{I}(x_i \notin b \wedge g(bin(x_i)) == g(b))s(bin(x_i))s(b) \tag{43}$$

Note that $E(\hat{n}) = n$. For varaince, observe that most of the terms in the summation have covariance 0, except the terms $Cov(\mathcal{I}(x_i \in b_1), \mathcal{I}(x_i \in b_2))$ which are negatively correlated. Hence

$$Var(\hat{n}) = \sum_{b,i}Var(\mathcal{I}(x_i \in b)) + Var(\mathcal{I}(x_i \notin b \wedge g(bin(x_i))! = g(b))s(bin(x_i))s(b)) +$$

$$2\sum_{i,b_1,b_2,b_1\ne b_2}Cov(\mathcal{I}(x_i \in b_1), \mathcal{I}(x_i \in b_2)) \tag{44}$$

We know that

$Var(\mathcal{I}(x_i \in b)) = p_b(1 - p_b)$

$Var(\mathcal{I}(x_i \notin b \wedge g(bin(x_i)) == g(b))s(bin(x_i))s(b)) = E(\mathcal{I}(x_i \notin b \wedge g(bin(x_i))! = g(b))^2) = \frac{1 - p_b}{R}$

$Cov(\mathcal{I}(x_i \in b_1), \mathcal{I}(x_i \in b_2)) = -p_{b_1}p_{b_2}$

Hence, we pluggin in the values in previous equation ,

$$Var(\hat{n}) = n \sum_b p_b(1 - p_b) + n \sum_b \frac{1 - p_b}{R} - 2n \sum_{b_1, b_2, b_1 \neq b_2} p_{b_1} p_{b_2} \tag{45}$$

$$Var(\hat{n}) = n(1 - \sum_b p_b^2) + n \sum_b \frac{1 - p_b}{R} - 2n \sum_{b_1, b_2} p_{b_1} p_{b_2} \tag{46}$$

$$Var(\hat{n}) = n\{(1 + \sum_b \frac{1 - p_b}{R} - (\sum_b p_b^2) - 2n \sum_{b_1, b_2} p_{b_1} p_{b_2})\} \tag{47}$$

$$Var(\hat{n}) = n\{(1 + \sum_b \frac{1 - p_b}{R} - (\sum_b p_b)^2\} \tag{48}$$

$$Var(\hat{n}) = n\{\sum_b \frac{1 - p_b}{R}\} \tag{49}$$

$$Var(\hat{n}) = \frac{n(n_{nzp} - 1)}{R} < \frac{n(n_{nzp})}{R} \tag{50}$$

Using Chebyshev's inequality , we have

$$P(|\hat{n} - n| > \epsilon n) \leq \frac{Var(\hat{n})}{\epsilon^2 n^2} \tag{51}$$

$$P(|\hat{n} - n| > \epsilon n) \leq \frac{n_{nzp}}{\epsilon^2 nR} \tag{52}$$

Hence with probability $(1 - \delta)$, $\delta = \frac{n_{nzp}}{\epsilon^2 nR}$, $\hat{n}$ is within $\epsilon$ multiplicative error.

**relation of pointwise Bias and ISB**    With probability $1 - \delta$,

$$\frac{\hat{f}_C(x)}{1 + \epsilon} \leq \hat{f}_C^*(x) \leq \frac{\hat{f}_C(x)}{1 - \epsilon} \tag{53}$$

As expectations respect inequalities

$$\frac{E(\hat{f}_C(x))}{1 + \epsilon} \leq E(\hat{f}_C^*(x)) \leq \frac{E(\hat{f}_C(x))}{1 - \epsilon} \tag{54}$$

$$\frac{E(\hat{f}_C(x))}{1 + \epsilon} - f(x) \leq Bias(\hat{f}_C^*(x)) \leq \frac{E(\hat{f}_C(x))}{1 - \epsilon} - f(x) \tag{55}$$

$$\frac{Bias(\hat{f}_C(x)) - \epsilon f(x)}{1 + \epsilon} \leq Bias(\hat{f}_C^*(x)) \leq \frac{Bias(\hat{f}_C(x)) + \epsilon f(x)}{1 - \epsilon} \tag{56}$$

$$\frac{Bias(\hat{f}_C(x)) - \epsilon f(x)}{1 + \epsilon} \leq Bias(\hat{f}_C^*(x)) \leq \frac{Bias(\hat{f}_C(x)) + \epsilon f(x)}{1 - \epsilon} \tag{57}$$

Integrating expressions again respects inequalities

$$\frac{ISB(\hat{f}_C(x)) - \epsilon \int f(x)}{1 + \epsilon} \leq ISB(\hat{f}_C^*(x)) \leq \frac{ISB(\hat{f}_C(x)) + \epsilon \int f(x)}{1 - \epsilon} \tag{58}$$

$$\frac{ISB(\hat{f}_C(x)) - \epsilon}{1 + \epsilon} \leq ISB(\hat{f}_C^*(x)) \leq \frac{ISB(\hat{f}_C(x)) + \epsilon}{1 - \epsilon} \tag{59}$$

Using first order taylor expansion of $\frac{1}{1+\epsilon}$ and ignore square terms

$$(1 - \epsilon)ISB(\hat{f}_C(x)) - \epsilon \leq ISB(\hat{f}_C^*(x)) \leq (1 + \epsilon)ISB(\hat{f}_C(x)) + \epsilon \tag{60}$$

$$ISB(\hat{f}_C(x)) - \epsilon(1 + ISB(\hat{f}_C(x))) \leq ISB(\hat{f}_C^*(x)) \leq ISB(\hat{f}_C(x)) + \epsilon(1 + ISB(\hat{f}_C(x))) \tag{61}$$

Hence,

$$ISB(\hat{f}_C(x)) - \epsilon N \leq ISB(\hat{f}_C^*(x)) \leq ISB(\hat{f}_C(x)) + \epsilon N \tag{62}$$

where

$$N = (1 + ISB(\hat{f}_C(x)))$$

**Point wise variance and IV**  Using the similar arguments

$$\frac{E(\hat{f}_C^2(x))}{(1+\epsilon)^2} - \frac{E^2(\hat{f}_C(x))}{(1-\epsilon)^2} \leq Var(\hat{f}_C^*(x)) \leq \frac{E(\hat{f}_C^2(x))}{(1-\epsilon)^2} - \frac{E^2(\hat{f}_C(x))}{(1+\epsilon)^2} \tag{63}$$

Again making first order taylor expansions of denominator and ignoring square terms

$$Var(\hat{f}_C(x)) - 2\epsilon(E(\hat{f}_C^2(x) + E^2(\hat{f}_C(x))) \leq Var(\hat{f}_C^*(x)) \leq Var(\hat{f}_C(x)) + 2(E(\hat{f}_C^2(x) + E^2(\hat{f}_C(x)))) \tag{64}$$

Since, $Var(\hat{f}_C(x)) = E(\hat{f}_C^2(x)) - E^2(\hat{f}_C(x))$

$$Var(\hat{f}_C(x)) - 2\epsilon(Var(\hat{f}_C(x)) + 2E^2(\hat{f}_C(x))) \leq Var(\hat{f}_C^*(x)) \leq Var(\hat{f}_C(x)) + 2\epsilon(Var(\hat{f}_C(x)) + 2E^2(\hat{f}_C(x))) \tag{65}$$

$$IV(\hat{f}_C(x)) - 2\epsilon(IV(\hat{f}_C(x)) + 2\int_{x\in S} E^2(\hat{f}_C(x))) \leq IV(\hat{f}_C^*(x)) \leq IV(\hat{f}_C(x)) + 2\epsilon(IV(\hat{f}_C(x)) + 2\int_{x\in S} E^2(\hat{f}_C(x))) \tag{66}$$

Let us now figure out the $\int_{x\in S} E^2(\hat{f}_C(x))$

$$\int_{x\in S} E^2(\hat{f}_C(x)) = \int_{x\in S} E^2(\hat{f}_H(x)) \tag{67}$$

From equation 18, $E(\hat{f}_H(x))^2 = f(x)^2 + (\langle(\frac{h}{2} - x), \nabla f(x)\rangle)^2 + 2f(x)\langle(\frac{h}{2} - x), \nabla f(x)\rangle$

$$\int_{x\in S} E^2(\hat{f}_H(x)) \leq \mathcal{G}(f) + \frac{h^2 d}{4}\mathcal{G}(\|\nabla f\|_2) + h\sqrt{d}\int_{x\in S}(f(x)\|\nabla f\|_2) \tag{68}$$

Hence,

$$IV(\hat{f}_C(x)) - 2\epsilon M \leq IV(\hat{f}_C^*(x)) \leq IV(\hat{f}_C(x)) + 2\epsilon M \tag{69}$$

Where

$$M \leq IV(\hat{f}_C(x)) + 2(\mathcal{G}(f) + \frac{h^2 d}{4}\mathcal{G}(\|\nabla f\|_2)) + h\sqrt{d}\int_{x\in S}(f(x)\|\nabla f\|_2)) \tag{70}$$

## A.5  LEMMA 1

Estimators $\hat{f}_S(x)$ and $\hat{f}_C^*(x)$, obtained from the Density Sketch with parameters(R,K,H) using histogram of width h built over n i.i.d samples drawn from true distribution have a relation

$$\int |\hat{f}_C^*(x) - \hat{f}_S(x)|dx = 2(1 - \rho_h)$$

where $\rho_h$ is the capture ratio as defined in section 3

$$\int |\hat{f}_C^*(x) - \hat{f}_S(x)|dx = \sum_{b\in bins}\int_{x\in b}|\hat{f}_C^*(x) - \hat{f}_S(x)|dx \tag{71}$$

$$\int |\hat{f}_C^*(x) - \hat{f}_S(x)|dx = \sum_{b\in bins(H)}\int_{x\in b}|\hat{f}_C^*(x) - \hat{f}_S(x)|dx + \sum_{b\notin bins(H)}\int_{x\in b}|\hat{f}_C^*(x) - \hat{f}_S(x)|dx \tag{72}$$

we know that for $x \in b, b \notin bins(H)$, $\hat{f}_S(x) = 0$. Hence,

$$\int |\hat{f}_C^*(x) - \hat{f}_S(x)|dx = \sum_{b\in bins(H)}\int_{x\in b}|\hat{f}_C^*(x) - \hat{f}_S(x)|dx + \sum_{b\notin bins(H)}\int_{x\in b}\hat{f}_C^*(x)dx \tag{73}$$

$\int_{x \in b} \hat{f}_C^*(x) dx$ is the probability of a data point lying in that bucket according to $\hat{f}_C^*(x)$

$$\int |\hat{f}_C^*(x) - \hat{f}_S(x)| dx = \sum_{b \in bins(H)} \int_{x \in b} |\hat{f}_C^*(x) - \hat{f}_S(x)| dx + \sum_{b \notin bins(H)} \frac{\hat{c}_b}{\hat{n}} \qquad (74)$$

For points $x \in b, b \in bins(H)$, $\hat{f}_C^*(x) * \hat{n} = \hat{f}_S(x) * \hat{n_h}$, Hence, $\hat{f}_S(x) = \frac{\hat{n}}{\hat{n_h}} \hat{f}_C^*(x)$

$$\int |\hat{f}_C^*(x) - \hat{f}_S(x)| dx = \sum_{b \in bins(H)} \int_{x \in b} \hat{f}_C^*(x) (\frac{\hat{n}}{\hat{n_h}} - 1) dx + \sum_{b \notin bins(H)} \frac{\hat{c}_b}{\hat{n}} \qquad (75)$$

$$\int |\hat{f}_C^*(x) - \hat{f}_S(x)| dx = \sum_{b \in bins(H)} \int_{x \in b} \hat{f}_C^*(x) (\frac{\hat{n}}{\hat{n_h}} - 1) dx + \sum_{b \notin bins(H)} \frac{\hat{c}_b}{\hat{n}} \qquad (76)$$

$$\int |\hat{f}_C^*(x) - \hat{f}_S(x)| dx = (\frac{\hat{n}}{\hat{n_h}} - 1) \sum_{b \in bins(H)} \frac{\hat{c}_b}{\hat{n}} + \sum_{b \notin bins(H)} \frac{\hat{c}_b}{\hat{n}} \qquad (77)$$

$$\int |\hat{f}_C^*(x) - \hat{f}_S(x)| dx = (\frac{\hat{n}}{\hat{n_h}} - 1)(\frac{\hat{n_h}}{\hat{n}}) + \frac{\hat{n} - \hat{n_h}}{\hat{n}} \qquad (78)$$

$$\int |\hat{f}_C^*(x) - \hat{f}_S(x)| dx = (1 - \frac{\hat{n_h}}{\hat{n}}) + \frac{\hat{n} - \hat{n_h}}{\hat{n}} \qquad (79)$$

$$\int |\hat{f}_C^*(x) - \hat{f}_S(x)| dx = 2(1 - \frac{\hat{n_h}}{\hat{n}}) \qquad (80)$$

$$\int |\hat{f}_C^*(x) - \hat{f}_S(x)| dx = 2(1 - \rho_h) \qquad (81)$$

## A.6 THEOREM 5

The IMSE of estimator $\hat{f}_S(x)$ obtained from the Density Sketch with parameters(R,K,H) using histogram of width h built over n i.i.d samples drawn from true distribution f(x) is

$$IMSE(\hat{f}_S(x)) \le 12(1 - \rho_h)^2 + 3IMSE(\hat{f}_C^*(x))$$

where $\rho_h$ is the capture ratio as defined in

*Proof.* Giving a very loose relation between $\hat{f}_S$ and f. We can write

$$\int (\hat{f}_S(x) - f(x))^2 dx = \int ((\hat{f}_S(x) - \hat{f}_C^*(x)) - (\hat{f}_C^*(x) - f(x)))^2 dx \qquad (82)$$

$$\int (\hat{f}_S(x) - f(x))^2 dx \le 3 \int (\hat{f}_S(x) - \hat{f}_C^*(x))^2 dx + 3 \int (\hat{f}_C^*(x) - f(x))^2 dx \qquad (83)$$

$$\int (\hat{f}_S(x) - f(x))^2 dx \le 3(\int |(\hat{f}_S(x) - \hat{f}_C^*(x))| dx)^2 + 3 \int (\hat{f}_C^*(x) - f(x))^2 dx \qquad (84)$$

$$\int (\hat{f}_S(x) - f(x))^2 dx \le 12(1 - \rho_h)^2 + 3 \int (\hat{f}_C^*(x) - f(x))^2 dx \qquad (85)$$

$$IMSE = MISE(\hat{f}_S(x)) \le 12(1 - \rho_h)^2 + 3IMSE(\hat{f}_C^*(x)) \qquad (86)$$

$\square$

## B   THEOREM 1 (MAIN THEOREM) COMBINES ALL OTHER THEOREMS

This theorem directly relates the distribution $\hat{f}_S(x)$ to the true distribution f(x). We will combine the following statements

$$\text{IMSE}(\hat{f}_H) \leq \frac{1}{nh^d} + \frac{\mathcal{G}(f)}{n} + o\left(\frac{1}{n}\right) + \frac{h^2 d}{4}\mathcal{G}(\|\nabla f\|_2) \tag{87}$$

$$\text{IMSE}(\hat{f}_C(x)) = \text{IMSE}(\hat{f}_H(x)) + \frac{n_{nzp}}{KRnh^d} \tag{88}$$

$$|\text{IMSE}(\hat{f}_C^*(x)) - \text{IMSE}(\hat{f}_C(x))| \leq \epsilon(N + 2M) \text{ with probability } (1 - \delta), \delta = \frac{n_{nzp}}{\epsilon^2 nR} \tag{89}$$

$$\text{IMSE}(\hat{f}_S) \leq 12(1 - \rho_h)^2 + 3\text{IMSE}(\hat{f}_C^*(x)) \tag{90}$$

where,

$$M \leq \text{IV}(\hat{f}_C(x)) + 2(\mathcal{G}(f) + \frac{h^2 d}{4}\mathcal{G}((\|\nabla f\|_2)) + h\sqrt{d}\int_{x \in S}(f(x)\|\nabla f\|_2))$$

$$N = (1 + \text{ISB}(\hat{f}_C(x)))$$

Let us now combine them

$$\text{IMSE}(\hat{f}_S(x)) \leq 12(1 - \rho_h)^2 + 3\text{IMSE}(\hat{f}_C^*(x)) \tag{91}$$

$$\text{IMSE}(\hat{f}_S(x)) \leq 12(1 - \rho_h)^2 + 3\left(\text{IMSE}(\hat{f}_C(x)) + \epsilon(N + 2M)\right) \tag{92}$$

$$\text{IMSE}(\hat{f}_S(x)) \leq 12(1 - \rho_h)^2 + 3\left(\text{IMSE}(\hat{f}_H) + \frac{n_{nzp} - 1}{KRnh^d} + \epsilon(N + 2M)\right) \tag{93}$$

$$\text{IMSE}(\hat{f}_S(x)) \leq 12(1 - \rho_h)^2 + 3\left(\frac{1}{nh^d} + \frac{\mathcal{G}(f)}{n} + o\left(\frac{1}{n}\right) + \frac{h^2 d}{4}\mathcal{G}(\|\nabla f\|_2)) + \frac{n_{nzp} - 1}{KRnh^d} + \epsilon(N + 2M)\right) \tag{94}$$

$$N = (1 + \text{ISB}(\hat{f}_C))$$

$$N \leq 1 + \frac{h^2 d}{4}\mathcal{G}(\|\nabla f\|_2)$$

$$M \leq \text{IV}(\hat{f}_C) + 2\mathcal{G}(f) + \frac{h^2 d}{4}\mathcal{G}(\|\nabla f\|_2) + h\sqrt{d}\int_{x \in S}(f(x)\|\nabla f\|_2)$$

$$M \leq \text{IV}(\hat{f}_H) + \frac{n_{nzp} - 1}{KRnh^d} + 2\mathcal{G}(f) + \frac{h^2 d}{4}\mathcal{G}(\|\nabla f\|_2) + h\sqrt{d}\int_{x \in S}(f(x)\|\nabla f\|_2)$$

$$M \leq \frac{1}{nh^d} + \frac{\mathcal{G}(f)}{n} + o\left(\frac{1}{n}\right) + \frac{n_{nzp} - 1}{KRnh^d} + 2\mathcal{G}(f) + \frac{h^2 d}{4}\mathcal{G}(\|\nabla f\|_2) + h\sqrt{d}\int_{x \in S}(f(x)\|\nabla f\|_2)$$

$$\text{IMSE}(\hat{f}_S(x)) \leq 12(1 - \rho_h)^2$$
$$+ 3\left(\frac{1}{nh^d} + \frac{\mathcal{G}(f)}{n} + o\left(\frac{1}{n}\right) + \frac{h^2 d}{4}\mathcal{G}(\|\nabla f\|_2)) + \frac{n_{nzp} - 1}{KRnh^d}\right)$$
$$+ 3\epsilon\left(1 + \frac{h^2 d}{4}\mathcal{G}(\|\nabla f\|_2) + \right.$$
$$\left. 2\left(\mathcal{G}(\|\nabla f\|_2) + \frac{1}{nh^d} + \frac{\mathcal{G}(f)}{n} + o\left(\frac{1}{n}\right) + \frac{n_{nzp} - 1}{KRnh^d} + 2\mathcal{G}(f) + \frac{h^2 d}{4}\mathcal{G}(\|\nabla f\|_2) + h\sqrt{d}\int_{x \in S}(f(x)\|\nabla f\|_2)dx\right)\right)$$

$$\text{IMSE}(\hat{f}_S(x)) \leq 12(1 - \rho_h)^2 +$$
$$3(1 + 2\epsilon)\left(\frac{1}{nh^d} + \frac{\mathcal{G}(f)}{n} + o\left(\frac{1}{n}\right) + \frac{n_{nzp} - 1}{KRnh^d}\right) +$$
$$3(1 + 3\epsilon)\frac{h^2 d}{4}\mathcal{G}(\|\nabla f\|_2)) +$$
$$3\epsilon(1 + 2\mathcal{G}(f) + h\sqrt{d}\int_{x \in S}(f(x)\|\nabla f\|_2))$$

.

## C  OTHER BASE LINES

*Coresets:* We considered a comparison with sophisticated data summaries such as coresets. Briefly, a coreset is a collection of (possibly weighted) points that can be used to estimate functions over the dataset. To use coresets to generate a synthetic dataset, we would need to estimate the KDE. Unfortunately, coresets for the KDE suffer from practical issues such as a large memory cost to construct the point set. Despite recent progress toward coresets in the streaming environment Phillips & Tai (2020), coresets remain difficult to implement for real-world KDE problems Charikar & Siminelakis (2017).

*Clustering and Importance Sampling:* Another reasonable strategy is to represent the dataset as a collection of weighted cluster centers, which may be used to compute the KDE and sample synthetic points. Unfortunately, algorithms such as $k$-means clustering are inappropriate for large streaming datasets and do not have the same mergeability properties as our sketch. Furthermore, such techniques are unlikely to substantially improve over random sampling when the samples is spread sufficiently well over the support of the distribution. An alternative approach is to select points from the dataset based on importance sampling Charikar & Siminelakis (2017), geometric properties Cortes & Scott (2016), and other sampling techniques Chen et al. (2012). However, recent experiments show that for many real-world datasets, random samples have competitive performance when compared to point sets obtained via importance sampling and cluster-based approaches Coleman & Shrivastava (2020).

*Dimensionality Reduction:* One can also apply sketching algorithms to compress a dataset by reducing the dimension of each data point via feature hashing, random projections or similar methods Achlioptas (2003). However, this is unlikely to perform well in our evaluation since our datasets are already relatively low-dimensional. Such algorithms also fail to address the streaming setting, where $N$ can grow very large, because the size of the compressed representation is linear in $N$. Finally, most dimensionality reduction algorithms do not easily permit the generation of more synthetic data in the original metric space.

## D  DIFFERENTIALLY PRIVATE DENSITY SKETCHES

In order to make the density sketch differentially private, we add noise to the distribution stored by density sketch. This is achieved by adding noise to the underlying count sketch array ($K \times R$ matrix of integers). Let the function mapping histogram of the data to the density sketch (before the heap construction) be denoted as $f : N^{|X|} \longrightarrow Z^{KR}$ where X is the set of all partitions. We fill first define an discrete analog of laplacian noise.

**Definition 1** (Double geometric distribution). *The double geometric distribution parameterized by* $p \in (0, 1)$ *is defined as follows on the support of all integers.*

$$P(z|p) = \frac{1}{2 - p}(1 - p)^{|z|}p \tag{95}$$

**Algorithm to make Density Sketches private:** Each cell of sketch ($K \times R$) matrix is added an i.i.d noise drawn from the double geometric distribution. We will prove that this noise addition makes the function $\mathcal{M} = f + noise$ differentially private. Heap construction can be considered as an post processing operation on the density sketch matrix. Hence, the sampling distribution is then

differentially private. (Note that heap construction algorithm also needs to be modified in practical settings to ensure that it carries the differential privacy properties. But this is achievable)

**Theorem 2** (Differential privacy)**.** *The density sketches constructed with addition of double geometric noise with $p = 1 - e^{-\epsilon/K}$ where K is the number of repetitions in the sketch is $(\epsilon, 0)$ differentially private.*

*Proof.* Consider the l1 metric for computing the distance between datasets. Consider any arbitrary pair x,y which satisfy $\|x - y\|_1 = 1$. In the histogram view of data, it is easy to check that a distance of 1 can exist if and only if there is an additional row in either $x$ or $y$ and all other data points are same. Without loss of generality we can write $x = y \cup \{d\}$ where $d$ is the extra data point. As the constructed count sketch does not depend on the order of insertion, we can say that count sketch for x, i.e. f(x), is obtained from count sketch for y by sketching additional data point into it. Also, because of countsketch's mergeable property, we can write $f(x) = f(y) + f(\{d\})$. Hence $\|f(x) - f(y)\|_1 = \|f(\{d\})\|_1$. As sketching a single entry changes exactly one element of each row of countsketch by 1. $\|f(\{d\})\|_1 = K$. Hence sensitivity of the function f is $\Delta f = K$

We use the double geometric distribution as defined above for noise.

$$P(z|p) = \frac{1}{2-p}(1-p)^{|z|}p \tag{96}$$

Now Let us consider the privacy achieved with this error. Let $\mathcal{M}$ be the final randomized algorithm with computation of f and adding noise. We are interested in the following quantity with x,y such that $\|x - y\|_1 = 1$.

$$\frac{P(\mathcal{M}(x) = z)}{P(\mathcal{M}(y) = z)} = \frac{\Pi_i P(\mathcal{M}(x)_i = z_i)}{\Pi_i P(\mathcal{M}(y)_i = z_i)}$$

$$= \frac{\Pi_i(1-p)^{|f(x)_i - z_i|}}{\Pi_i(1-p)^{|f(y)_i - z_i|}}$$

$$= \Pi_i(1-p)^{|f(x)_i - z_i| - |f(y)_i - z_i|}$$

$$= (1-p)^{\|f(x)-z\|_1 - \|f(y)-z\|_1}$$

As l1-norm is a distance metric we can write

$$\frac{P(\mathcal{M}(x) = z)}{P(\mathcal{M}(y) = z)} = (1-p)^{\|f(x)-z\|_1 - \|f(y)-z\|_1}$$

$$\geq (1-p)^{\|f(x)-f(y)\|_1}$$

$$= (1-p)^{\Delta(f)}$$

If we put $p = 1 - e^{-\epsilon/\Delta(f)}$

$$\frac{P(\mathcal{M}(x) = z)}{P(\mathcal{M}(y) = z)} \geq e^{-\epsilon}$$

$$\frac{P(\mathcal{M}(y) = z)}{P(\mathcal{M}(x) = z)} \leq e^{\epsilon} \tag{97}$$

Hence $\mathcal{M}(x)$ is $(\epsilon, 0)$- differentially private. Hence we have that the countsketch produced by the sketching algorithm with added double geometric noise is $(\epsilon, 0)$- differentially private when we have $p = 1 - e^{-\epsilon/K}$

**why heaps are differentially private?** If the data is bounded in $R^d$ (d is the dimension of the data), then it is easy to check that there is a cell in $R^d$, which contains all the data, It follows that the number of partitions inside this cell is finite. So we can consider heap construction as iteratively going through each partition and noting down its count. Once we do that, we sort all the partitions according to the counts and keep top H elements. In this sense, we can consider heap construction as a post processing over count sketch. From the proposition 2.1 [Dwork, Roth], we know that post processing maintains differential privacy. Hence the heap we create is $(\epsilon, 0)$ differentially private □

