# OpenReview forum: "Density Sketches for Sampling and Estimation"
_ICLR.cc/2023/Conference — Submitted to ICLR 2023_

### Official Review · Reviewer_VE18 · 2022-10-25

**Confidence:** 2
**Clarity, Quality, Novelty And Reproducibility:** Mostly well written. Some interesting…
**Correctness:** 3
**Technical Novelty And Significance:** 2
**Empirical Novelty And Significance:** Not applicable
**Recommendation:** 5

**Strength And Weaknesses:**

Pros: Interesting algorithmic ideas that work for low-dimensional, Euclidean data.
Cons: Most modern ML problems are high-dimensional problems that do not involve Euclidean data. In such cases density estimation is done via powerful parametric models. For problems which have a Euclidean structure, density estimation procedures cannot scale beyond a few dimensions.

**Summary Of The Paper:**

The paper presents density sketches, which is a cheap and practical way to reduce data in a streaming setting., DS can keep a succint representation of data and can sample unseen data on the fly from this succint represesentation. This is done by reducing KDE into a problem related to histograms. The integrated mean squared error is analyzed. Experimental analysis is perofmred on small scale datasets.

**Summary Of The Review:**

NIce algorithmic contributions to density estimation literature. But, these approaches will not sale to larger dimensionality, Euliean data.

---

### Official Review · Reviewer_Kdqf · 2022-10-28

**Confidence:** 2
**Correctness:** 3
**Technical Novelty And Significance:** 2
**Empirical Novelty And Significance:** 2
**Recommendation:** 5

**Clarity, Quality, Novelty And Reproducibility:**

While the paper is well-organized, it is very hard to read due to so many notation problems and typos. See my comments in the summary.

**Strength And Weaknesses:**

The method is simple and seems effective from the empirical evaluation. The paper is well-organized and easy to follow.

**Summary Of The Paper:**

This paper proposes an approach to estimate the data distribution. For high-dimensional spaces, the traditional histogram-based algorithm and KDE methods require #data exponential in the dimension. The paper proposes Density Sketch (DS) which uses particular data structure Count Sketch (CS) to reduce the memory consumption of histogram-based methods. Moreover, the paper proposes to use a min-heap for efficient sampling from the estimated distribution.

**Summary Of The Review:**

While the paper is well-organized, it is very hard to read due to so many notation problems and typos. Here are my questions:

1. In many places in the paper, what is the dot . in the denominator of $\hat f_H$? (i.e., the $n.V(B(x))$)

2. What is $\hat C(x)$ in Section 4.4? What is the difference between Section 4.3 and 4.4? I don't understand why we don't just use $\hat f^*_C(x)$ for estimating the density. As stated in the paper, $\hat f_C(x)$ is not normalized ...

3. In Section 5, what is the probability space of the $E$ in MISE? And what is exactly IMSE? Is there a typo in the second equation in page 7? What is the randomness in Theorem 1 ("with probability ($1-\delta$)"). The argument in the analysis is that `if MISE -> 0, then $\hat f$ converge to $f$ in probability'. Does the randomness affect this result or not?

4. The author should give more background on L1/L2 LSH in the maintext as they are mentioned a lot later.

5. There are many notation problems in the paper. For example, in Section 4.2 when introducing notations, what is $f(x): S \subset R^d \rightarrow R$ intend to tell us? My guess is $S$ a Borel set, but not sure .... Also, the author simply writes $p$ for p as well as many other symbols in many places in the paper. The author seems submitted this work in a rush and there are even paragraphs without period.

6. While the author claims that DS has advantages for high-dimensional spaces, it seems that the evaluations are still performed on datasets with dimensions $<100$.

I suggest the author do a thorough proofreading of the paper to improve the paper's readability and clarity.

---

> ### Author Response · Authors · 2022-11-16
> **Response to questions**
>
> We thank the reviewer for their thoughtful comments and questions. We have fixed the mathematical typographical issues in the newer version.
> (point numbers correspond to point numbers above )
>
> 2. $\hat{C}(x)$ is the estimate of the count of elements in bin(x)  from the count sketch.  As pointed out by the reviewer, $\hat{f}^*_C(x)$ can be used to perform point-wise density estimation (density at a point) as well. However $\hat{f}_C(x)$ is a better estimate for point-wise density. As mentioned in section 4.3-4.4, $\hat{f}_C(x)$ is not an estimate of $f(x)$ but $\hat{f}_C^*(x)$ is.
> Hence, we propose $\hat{f}_C(x)$ for point-wise estimates and $\hat{f}^*_C(x)$ for estimating $f(x)$.
>
> 3. IMSE and MISE are standard tools used in density estimation analysis. They are briefly defined in section 5. There was a typo in the equation. It is IMSE = MISE.
>
> The randomness arises due to two factors. First, due to the data used to estimate $\hat{f}(x)$ being a random sample drawn from the true distribution $f(x)$. Secondly, the random hash functions are drawn from a set of universal hash functions. The expectation is over both these random choices.  The randomness specific to $(1 - \delta)$ arises due to random hash functions. This randomness does not affect the argument about convergence in probability. This randomness will get absorbed into the final probabilistic statement and we still have convergence in probability.
>
> 5. S is the support of the $f(x)$. Practically we consider S is the bounding box of support in $R^d$.
>
> 6. Density estimation in high dimensions (including that performed by DS ) require exponentially larger datasets. While increased data collection is combating curse-of-dimensionality, such large datasets are not available in public domain.  Hence, we end up choosing datasets with $d < 100$. DS will work well with higher dimensions if enough data is available.

---

### Official Review · Reviewer_jjJ4 · 2022-11-03

**Confidence:** 4
**Correctness:** 3
**Technical Novelty And Significance:** 3
**Empirical Novelty And Significance:** 3
**Recommendation:** 6

**Clarity, Quality, Novelty And Reproducibility:**

The paper needs a good polish in its mathematical expressions and language. Another weird point is that the paper was uploaded in non-searchable form (I had to OCR It). It would be very useful for reviewing to have at least the text searchable, and ideally all references hyperlinked.

The combination of a heavy-hitter list along with LSH bins and count sketching is novel to me, but there have been many streaming data reduction techniques and it is difficulty to fully judge the novelty without being directly in this research area. Code was provided, and the experiments appear reproducible.

If, after discussions, the main result is determined to truly represent an approximation theorem in the limit, then there is enough contribution here to make it worthy to share with the community, conditionally on clarifying the presentation and polishing the typesetting and language.


**Strength And Weaknesses:**

### Strengths

* Reducing the overhead of data by representing it accurately in compressed form continues to be a timely topic.

* The construction of the density sketch data structure is intuitive and the compression and accuracy levels achieved in experiments seem very promising.

### Weaknesses

* The paper is typographically in a very rough form. Many mathematical expressions are not placed in math style and there are typesetting errors in those in math style (e.g., hat symbols appearing in the wrong places) that make expressions hard to parse. There are also many linguistic issues, some of which are listed under minor comments. The paper needs a serious polish revision to become of publication quality.

* The main theorem (Theorem 1) is not presented cleanly. Here are some issues:
    * The parentheses are all the same size, making it hard to parse what goes where. This is an easy fix, with the exception that there seems to be an extra right parenthesis in the very end, which could change the meaning of the expression if it is associated with a left parenthesis somewhere.
    * Perhaps a related issue to the above, but as written the right-hand side does *not* vanish, even if both $1/nB^d$ and $B^2d$ vanish. That’s because the last parenthesis block which starts with $3\epsilon(1+2\mathcal{G}(f)+\cdots)$ is not affected by either of these. This could be an issue of misplaced parentheses. However, as far as I can tell from the proof (which I have only skimmed through), that term is there.

* The reasons the last point necessitates a clarification is because it makes the conclusions from the Theorem a bit more subtle. In particular, it would seem that we can make $\delta$ a constant, which implies that $\epsilon$ vanishes, so this term would vanish. However, if I understand it correctly from the proof, the equation is only valid if $\delta$ is small for a fixed $\epsilon$. Therefore we cannot change the order of quantifiers, and that reasoning does not work. I welcome engaging with the authors to clarify this point, as going through the proof to reach the conclusion myself is a bit daunting.

* The paper would benefit from a clear characterization of the worst-case space and time complexity of the method, since lower complexity is one of the selling points.

### Questions

* In Figure 1, what is $g_i(a)$?
* In SRP, are the volumes of all partitions equal? You assume that throughout, but perhaps that only works for LSH?
* I don’t think you mean multinomial distribution in Algorithm 3, you instead mean a categorical distribution over the bins.



### Typos and suggestions

(p.1) include edge > including edge
(p.2) in the section 2 > in this section. ; function f(x) > function $f(x)$
(p.3) place $\hat$ symbols correctly; define $\mathcal{C}$ as the count vector
(p.4)
* Your use of $i$ is too overloaded, you use it to index: data points, coordinates, and LSH/SRP weights. It would be more legible if each of theses uses had its own letter.
* In Table 1:
   * for LSH and SRP in the bin column, there shouldn’t be a subscript on $x$
   * typeset the $w_1$ vector and $W$ vector in math mode
   * remove trailing parenthesis after $U[0,1]$
* point x > point $x$
(p.5) a efficient > an efficient; in the figure > in Figure; fix all the $\hat$ locations (last mention, this has to be fixed everywhere in the paper); remove trailing parenthesis from $\hat \mathcal{C}(x)$
(p.6) in Algorithm 1, fix the partition function line; in Algorithm 3, remove trailing parenthesis from UniformRandomPoint(b); point x > point $x$; close parenthesis after MSE
(p.7) add = between IMSE and MISE


**Summary Of The Paper:**

This paper constructs a data structure, dubbed density sketch, that represents a density on a high-dimensional space. This data structure is constructed online, in a streaming fashion from data samples assumed drawn i.i.d. from a true density. The representation is capable of evaluating the density at specific points as well as of sampling points. The paper also provides analysis of the approximation level of this representation in MISE distance and illustrates its applicability via some experiments.

**Summary Of The Review:**

The paper puts together various data reduction techniques in an intuitive manner to offer a novel density representation that can be constructed in a streaming fashion. This appears to lead to low space and time complexity, but ideally that analysis should also be part of the paper. The main result requires a bit of clarification, to make sure that the approximation ability of this representation is indeed there. What hampers the paper most is unpolished typesetting and language. With some clarification and polish, the paper may be worth sharing with the community.

---

> ### Author Response · Authors · 2022-11-11
> **Main theorem clarification**
>
> We thank the reviewer for pointing out the typographical errors and we have fixed most of the issues in current revision. (We are working on full fixes but wanted to start the discussion on main theorem as soon as possible. The current version should help do that). We appreciate the effort put by the reviewer in understanding our paper. Looking forward to the discussion.
>
> We updated the main theorem with correct bracketing and its interpretation. Specific to the question on $\epsilon, \delta$ see the comment 2 below. Also, to get a little in-depth understanding of the theorem, you can refer to equations (87-90) in appendix B (updated appendix is attached to the main paper itself) which show the statements of IMSE on individual approximation steps.
>
> Notation help for comments below : $B$ - bin-width, $n$ number of data points, $n_{nzp}$ number of non-zero partitions $\epsilon, \delta$ standard trade-off parameters of count sketch. $K,R$ repetition and range parameters of count sketch. $\rho_h$ capture ratio measures amount of data captured in heap.
>
> 1. Similar to the standard analysis for histograms, the curse of dimensionality manifests in our theorem as well. $B$ should go to zero and $n$ should increase faster than the rate at which $B^d/n_{nzp}$ decreases (condition 1). As compared to standard histograms, this requires $n$ to grow faster. With these conditions on $B$ and $n$ it is clear how second and third terms go to zero.
>
> 2. The magnitude of fourth term is controlled via $\epsilon$. The main statement is true for any $\delta$ and $\epsilon$ that are related via the expression $\delta = \left(n_{nzp} / (\epsilon^2 n K R)\right)$ and $\delta < 1$ . Choose arbitrarily small $\epsilon$ and $\delta$, we can achieve it with large enough $n/n_{nzp}$ or by providing more intermediate resources and making $KR$ large enough. For a fixed resource $KR$, this term goes to zero asymptotically with $n$ growing faster than $n_{nzp}$ which is a sub-condition of condition 1.
>
> 3. The term $12(1-\rho_h)^2$ shows the effect of truncation that occurs due to using only heavy partitions. As can be seen, this term is data dependent and IMSE does not depend directly on $H$ (number of partitions in heap) but $\rho_h$. If we can capture the entire data in heap (i.e setting $H = n_{nzp}$), then the term adds no penalty to IMSE. $H$, via $\rho_h$ controls the accuracy-memory trade-off of DS.

---

> ### Author Response · Authors · 2022-11-12
> **Other comments/questions**
>
> We thank the reviewer for pointing out issues with the format in which the paper was uploaded. Our current version is uploaded in a searchable format with references hyperlinked. We comment on the rest of the questions in this comment (for the main theorem discussion, we have initiated a separate comment).
>
> 1. Complexities:  time-complexity to process each sample $O(d^2 + Kd)$, Space complexity of DS during construction: $O(KR + Hd)$. The time complexity of sampling from L2-LSH DS is $O(d^3)$. The space complexity of DS during sampling. $O(Hd)$
>
> 2. Questions.
>
>    a. $g_i(a)$ is the sign hash function of the count-sketch. It maps the key to a range {+1, -1}
>
>    b.  We assume regular partitions in our analysis. SRP will not have regular partitions. However, one may choose to use it in practice as it is also a form of a randomized histogram and can be used for sampling.
>
>    c. Corrected in a new version (will be uploaded soon)
>
> (We have uploaded the current version with fixes for math typographical issues in the main paper and most parts of the appendix. We are working on complete fixes and will be uploaded soon)

---

### Decision · Program_Chairs · 2023-01-20

**Decision:**

Reject

**Justification For Why Not Higher Score:**

The paper is too hard to read, even after the revision.

**Justification For Why Not Lower Score:**

n/a

**Metareview: Summary, Strengths And Weaknesses:**

The paper proposes a density sketch for a data stream aimed to represent the data distribution. From the reviews and discussion, the paper has several strengths: The goal of obtaining a compressed representation of the data is well motivated. The technique provided is sensible and in high level, the paper provides proper intuition for it. Finally, the empirical results show a promising increase over the baseline.
The main weakness of the paper, raised in the reviews is the quality of its writing. The original version had many typos and confusing notations making it hard to verify the correctness of the arguments. A major issue related to this is the main theorem, that is quite difficult to comprehend. The authors response and new version mitigated some of these issues, but not all of them. Furthermore, the scope of the problem was too large to be fixed in a rebuttal period, and a new version should be thoroughly reviewed. Concluding, in its current form, the paper is not ready for publication.